# Antitumor and Antioxidant Activities of In Vitro Cultivated and Wild-Growing *Clinopodium vulgare* L. Plants

**DOI:** 10.3390/plants12081591

**Published:** 2023-04-09

**Authors:** Maria Petrova, Lyudmila Dimitrova, Margarita Dimitrova, Petko Denev, Desislava Teneva, Ani Georgieva, Polina Petkova-Kirova, Maria Lazarova, Krasimira Tasheva

**Affiliations:** 1Department of Plant Ecophysiology, Institute of Plant Physiology and Genetics, Bulgarian Academy of Sciences, Acad. G. Bonchev Str., 21, 1113 Sofia, Bulgaria; marry_petrova@yahoo.com (M.P.); dim.lyudmila@gmail.com (L.D.); mstoyadinova@abv.bg (M.D.); 2Laboratory of Biologically Active Substances, Institute of Organic Chemistry with Centre of Phytochemistry, Bulgarian Academy of Sciences, 4000 Plovdiv, Bulgaria; petko.denev@orgchm.bas.bg (P.D.); desislava.teneva@orgchm.bas.bg (D.T.); 3Department of Pathology, Institute of Experimental Morphology, Pathology and Anthropology with Museum, Bulgarian Academy of Sciences, 1113 Sofia, Bulgaria; georgieva_any@abv.bg; 4Department of Synaptic Signaling and Communication, Institute of Neurobiology, Bulgarian Academy of Sciences, 1113 Sofia, Bulgaria; kirovaps@yahoo.com (P.P.-K.); m.lazarova@gmail.com (M.L.)

**Keywords:** wild basil, micropropagation, polyphenols, HPLC, antitumor activity

## Abstract

*Clinopodium vulgare* L. is a valuable medicinal plant used for its anti-inflammatory, antibacterial and wound-healing properties. The present study describes an efficient protocol for the micropropagation of *C. vulgare* and compares, for the first time, the chemical content and composition and antitumor and antioxidant activities of extracts from in vitro cultivated and wild-growing plants. The best nutrient medium was found to be Murashige and Skoog (MS) supplemented with 1 mg/L BAP and 0.1 IBA mg/L, yielding on average 6.9 shoots per nodal segment. Flower aqueous extracts from in vitro plants had higher total polyphenol content (29,927.6 ± 592.1 mg/100 g vs. 27,292.8 ± 85.3 mg/100 g) and ORAC antioxidant activity (7281.3 ± 82.9 µmol TE/g vs. 7246.3 ± 62.4 µmol TE/g) compared to the flowers of wild plants. HPLC detected qualitative and quantitative differences in phenolic constituents between the in vitro cultivated and wild-growing plants’ extracts. Rosmarinic acid was the major phenolic constituent, being accumulated mainly in leaves, while neochlorogenic acid was a major compound in the flowers of cultivated plants. Catechin was found only in cultivated plants, but not in wild plants or cultivated plants’ stems. Aqueous extracts of both cultivated and wild plants showed significant in vitro antitumor activity against human HeLa (cervical adenocarcinoma), HT-29 (colorectal adenocarcinoma) and MCF-7 (breast cancer) cell lines. The best cytotoxic activity against most of the cancer cell lines, combined with the least detrimental effects on a non-tumor human keratinocyte cell line (HaCaT), was shown by the leaf (250 µg/mL) and flower (500 µg/mL) extracts of cultivated plants, making cultivated plants a valuable source of bioactive compounds and a suitable candidate for anticancer therapy.

## 1. Introduction

Medicinal plants have been used for centuries to treat a number of diseases and represent an important source for identifying and obtaining novel pharmaceutical drugs [1]. Scientific evidence for the benefits of using medicinal plants and plant-derived active compounds as safe and non-toxic remedies is steadily growing. Due to their wide range of biological activities and favorable effects, plants of the Lamiaceae family are particularly suitable as a source of new phytochemicals. *Clinopodium vulgare* L. (wild basil), spread throughout most of Europe, Western and Central Asia, North America and Northern Africa, is an aromatic herb belonging to the Lamiaceae family. In Bulgaria, the species is used in traditional medicine for the treatment of wounds, warts, prostatitis, mastitis, diabetes and stomach ulcers [2,3]. It is officially recognized as a herb with medicinal properties under the Medicinal Plants Act, Bulgaria [4], and also as a functional food, promoting health and helping reduce the risk of disease, by the International Food Information Council Foundation (2011) [5]. The species is especially rich in biologically active compounds such as polyphenols, flavonoids, triterpenes, triterpenoid saponins, phenolcarboxylic acids, monoterpenoids, diterpenoid quinones and triterpenoid glycosides, mainly found in the aerial parts of the plant [6,7,8,9,10]. Hence, it has been shown that it holds anti-inflammatory, DNA-protective and antibacterial properties [10,11] and has been found to possess high antioxidant and enzyme-inhibiting activities [3].

Conventional plant breeding methods, as well as plant tissue culture, can change and, on many occasions, improve the yield and agronomic and medicinal traits of plants [12]. Biotechnological approaches provide the opportunity to obtain high-quality clones and overcome the variability in wild-growing plants (depending on the conditions of the specific habitat, local climate and geographical features), eliminate the existence of toxic components and contaminants, increase the contents of biologically active compounds and improve beneficial medicinal effects [13]. Information on the in vitro propagation of *Clinopodium* spp. is limited mainly to *C. odorum* [14,15,16] and *C. nepeta* [17]. There is only one report mentioning the tissue culture of *C. vulgare* [18], which, however, does not provide information on the chemical composition of in vitro cultivated plants, nor give a comparison with the wild-growing population. Furthermore, studies investigating the antitumor activity of *C. vulgare* are scarce, and they refer only to wild plants [19]. It was found that extracts from wild *C. vulgare* plants possess cytotoxic activity against several cell cancer cell lines, such as A2058 (human metastatic melanoma), HEp-2 (human larynx epidermoid carcinoma) and L5178Y (mouse lymphoma), as well as CaOV (human testis cystadenocarcinoma), HeLa (human cervical adenocarcinoma) and HT-29 (human colorectal adenocarcinoma). To our knowledge, the antitumor activity of in vitro cultivated *C. vulgare* plants has not been investigated so far. Therefore, the aim of the present work was to develop an efficient protocol for the in vitro propagation of *C. vulgare* plants and to compare their chemical content and composition and antioxidant and antitumor activities with those of wild-growing plants.

## 2. Results

### 2.1. In Vitro Culture

#### 2.1.1. Shoot Induction and Multiplication 

The seed surface sterilization treatment provided 100% decontaminated seeds and was successful in ensuring an axenic culture. The results showed that seed germination on Murashige and Skoog [20] basal medium (MS) free of plant growth regulators (PGRs) was high. The achieved germination rate was 95% on the 21st day. The shoot tips and nodal segments from two-month-old in vitro seedlings showed high regeneration potential and were suitable for starting the micropropagation of *C. vulgare*. Shoot initiation occurred five to six days after inoculation on all examined nutrient media. The addition of PGRs to the nutrient media significantly increased shoot production compared to the control (PGR-free) treatment. The explants cultivated on the control medium grew in height, but only 10% developed new shoots within the tested period. It was found that the type and concentration of the plant growth regulator and the type of initial explant affected the frequency and rate of multiplication (Table 1, Figure 1). Nodal segments were more efficient in producing new shoots compared to shoot tips. Among the examined cytokinins, 6-benzylaminopurine (BAP) was the most effective regarding shoot initiation (an average value of 2.05 per apical explant and an average value of 4.6 shoots per node). Shoot proliferation and the shoot number per explant were lower in N6 [2-isopentenyl]-adenine (2-iP)-containing medium. Nutrient media supplemented with both cytokinins and auxins were more effective for shoot multiplication compared to those containing only cytokinins. The best results were obtained on MS medium enriched with 1 mg/L BAP and 0.1 mg/L indole-3-butyric acid (IBA), in which 100% of the nodal segments developed shoots, and the multiplication rate reached 6.9 shoots per explant (Table 1, Figure 2a). Supplementation with zeatin and IBA and the combination of kinetin and IBA also promoted multiplication, and the mean number of shoots per nodal explant was 4.0 and 3.6, respectively. The medium containing 2-iP and IBA was less effective for shoot proliferation. The explants maintained on MS medium free of PGRs showed the best results regarding the height of shoots (8.3 cm). When kinetin was added to the nutrient medium, the shoots’ height increased as well. The multiplication rate increased in the second subculture in the optimal nutrient medium. The mean number of shoots per explant reached 7.8 for nodal segments and 4.2 for shoot tip explants. Decreases in the multiplication rate and regenerative potential of *C. vulgare* were observed in the subsequent subcultures. The shoots grew in height (9–12 cm) within the 4-week culture period on all tested media, were cut into 3–4 nodal segments and again subcultivated on fresh media, resulting in a large number of new shoots.

#### 2.1.2. In Vitro Rooting and Acclimation

In vitro shoots were successfully rooted (100%) on half-strength MS medium free of auxins (Figure 2b). This low-cost medium ensured the induction of rhizogenesis and the obtainment of a well-developed root system suitable for subsequent ex vitro adaptation. The mean number of roots per explant was 6.7 with a mean length of 5.25 cm. The substrate mixture consisting of peat, perlite, sand and soil (2:1:1:1) was appropriate for the ex vitro adaptation of the in vitro rooted plants and for plants’ hardening (the survival rate was 100%). No phenotypic variations were observed in the ex vitro adapted plants (Figure 2c). The plants were successfully acclimatized to the experimental field (“Elin Pelin” at 640 m a. s. l.), and most of the plants (90%) bloomed in the first year.

### 2.2. Phytochemical Analyses

#### 2.2.1. Total Polyphenol and Total Flavonoid Contents

Total polyphenol and flavonoid contents as well as the antioxidant activity of in vitro cultivated and wild plants’ extracts are shown in Table 2. The amount of polyphenols varied between 16,713.6 mg GAE/100 g DW and 30,510.9 mg GAE/100 g DW. The highest total polyphenol content was found in leaf extracts from wild plants, followed by the flower extract of in vitro cultivated plants. More polyphenols were measured in the flower extracts of in vitro plants in comparison to those of wild plants. No differences in polyphenol content were observed between stem samples. The highest content of flavonoids was reported in leaf extracts, with the trend holding for both in vitro and wild plants. Extracts from wild-growing plants were richer in flavonoids compared to the samples of in vitro derived plants.

#### 2.2.2. Antioxidant Activity 

The extract from the leaves of wild-growing plants exhibited the highest activity as assessed by HORAC (2572.7 µmol GAE/g), followed by the extracts of flowers of wild and in vitro cultivated plants. The detected ORAC activity was not significantly different between the flower and the leaf extracts from both in vitro cultivated and wild-growing plants. The lowest activity as assessed by ORAC was observed in the extract from stem samples.

#### 2.2.3. HPLC Analysis

HPLC analysis detected qualitative and quantitative differences in the phenolic constituents between the in vitro cultivated and wild-growing plants’ extracts (Table 3). Rosmarinic acid was the most abundant phenolic compound in the samples from both in vitro cultivated and wild-growing plants, and leaf extracts contained the highest amount of rosmarinic acid, followed by flower extracts. Catechin was detected only in the flowers and leaves of in vitro cultivated plants, whereas quercetin and apigenin were found only in stem extracts, predominantly in the in vitro plants. Caffeic acid was high in the flowers of both in vitro cultivated and wild plants. Neochlorogenic acid was high in the flowers of in vitro cultivated plants.

### 2.3. Antitumor Activity

#### 2.3.1. MTT Cell Viability Assay

The antitumor activity of *C. vulgare* extracts obtained from in vitro grown and wild plants was evaluated on three human tumor cell lines—HeLa human cervical adenocarcinoma, HT-29 human colorectal adenocarcinoma and MCF-7 breast cancer—after 24 and 48 h of exposure using the MTT test. The effects of the extracts on the viability of the non-tumor human keratinocyte cell line HaCaT was also examined to assess the selectivity of the cytotoxic action with respect to tumor cells. All *C. vulgare* extracts induced a concentration- and time-dependent reduction in the cell viability and proliferation of the tested cell lines (Figure 3).

The extracts obtained from leaves were the most active, followed by flower and stem extracts. The extracts from the respective organs of in vitro grown and wild plants showed comparable cytotoxic activity. Moreover, in the HT-29 cell line, LCP applied at concentrations of 250 µg/mL and 500 µg/mL showed significantly higher cytotoxic activity than LWP. The cell viability of colorectal carcinoma cells treated with 250 µg/mL LCP for 24 and 48 h was reduced to 70.8% and 57.2%, respectively, while the same concentration of LWP did not induce a significant reduction in cell viability. The treatment of HT-29 cells with 500 µg/mL LCP and LWP reduced the cell viability to 9.3% and 12.2%, respectively, at the 24th hour and to 5.4% and 7.6% at the 48th hour. The flower extracts showed the highest selectivity for tumor cells. The viability of all tumor cell lines treated with 500 µg/mL FCP and FWP was significantly lower than the viability of the non-tumor HaCaT cells exposed to the same concentrations of the flower extracts. On the basis of the MTT assay results, the half-maximal inhibitory concentrations (IC_50_) were calculated for all extracts and cell lines tested (Table 4).

The IC_50_ values of the tested extracts ranged from 266.5 µg/mL to 934.1 µg/mL. The lowest IC_50_ value was established in the HT-29 cell line treated for 48 h with an extract from the leaves of in vitro cultivated plants. The presented results indicate that the colorectal carcinoma cell line HT-29 showed the highest sensitivity.

#### 2.3.2. Fluorescence Microscopy

To analyze the mechanisms of the antitumor activity, HT-29 colorectal carcinoma cells treated for 24 h with an extract obtained from the flowers of in vitro cultivated *C. vulgare* were examined by fluorescence microscopy after staining with acridine orange/ethidium bromide (AO/EB) and 4′,6-diamidino-2-phenylindole (DAPI). The tested *C. vulgare* extract induced marked alterations in the cellular and nuclear morphology of the tumor cells (Figure 4). 

Control HT-29 cells stained with AO/EB showed typical morphology and growth characteristics and homogeneous green staining (Figure 4a). The changes in cell cultures exposed to the *C. vulgare* extract were indicative of both an antiproliferative effect and the induction of cell death. The cell density was significantly reduced, and a large number of early apoptotic cells with intense green or yellow fluorescence, as well as red-stained late apoptotic cells, were present (Figure 4b).

To confirm the ability of the tested extract to induce apoptotic cell death, we analyzed the nuclear morphology of the tumor cell after DAPI staining. The nuclei of untreated cells had an oval shape, smooth edges and homogeneous blue staining. Tumor cells formed a monolayer, and numerous mitotic cells were observed (Figure 4c). After treatment with the *C. vulgare* extract, cell nuclei were shrunken with uneven outlines, and cells in mitosis were not found. The chromatin was condensed and showed intense fluorescent staining that is characteristic of apoptotic cell death (Figure 4d).

## 3. Discussion

In this study, a simple and efficient micropropagation protocol using shoot tips and nodal segments from in vitro raised seedlings was developed. The seeds collected from wild-growing plants were successfully used for the establishment of an in vitro culture. The observed high germination rate (95%) is in line with the conclusion of Vlachou et al. [17] that most *Clinopodium* species do not enter dormancy after seed harvest and do not need any pretreatment. The type of explant, the chemical composition of the main nutrient medium and the PGRs used are factors that influence the regeneration potential of plants grown under in vitro conditions [21]. Several studies have reported the effect of nutrient media with different compositions of macro- and micro-salts on plant tissue cultures of *Clinopodium* species. *C. odorum* was successfully propagated on Woody plant medium without the use of growth regulators [14]; however, higher concentrations of pulegone (a volatile organic compound) were accumulated on half-strength salt media (MS and B5 Gamborg medium) [16]. MS medium was applied for slow growth maintenance within the 12-month culture period of *C. vulgare* [18]. In our study, the MS basal medium was successfully applied, but the addition of cytokinins was necessary to stimulate shoot multiplication. It is well known that cytokinins induce adventitious shoots by promoting plant cell division, contributing to DNA synthesis and controlling the cell cycle [22]. Each of the cytokinins tested, BAP, kinetin, zeatin and 2-iP, had a significant effect on the number of shoots per explant. The maximum multiplication of *C. vulgare* was observed on the medium containing BAP. The efficiency of BAP in inducing shoot growth and proliferation may be due to the plant’s ability to metabolize it more readily compared to other synthetic growth regulators. Other reasons could be its lower resistance to cytokinin oxidase, easy permeability and induction of the synthesis of natural plant hormones, such as zeatin [23,24]. Many studies have shown that combined cytokinins and auxins have a synergistic effect on shoot multiplication, possibly because auxins have been found to regulate the size of the pool of active cytokinins by stimulating their glucosylation and oxidative breakdown [25]. For example, it has been demonstrated that, in *C. nepeta*, the addition of BAP (8 mg/L) and NAA (0.1 mg/L) to the MS medium resulted in the elimination of hyperhydricity and induced an increase in shoot production of 80–88%, with seedlings yielding 6.5 shoots per explant and adult-origin explants yielding 7.5 shoots per explant [17]. In our study with *C. vulgare*, 1 mg/L BAP was already enough to yield a multiplication frequency of 100% and a mean number of 6.9 shoots per nodal segment on MS medium. The most reliable and widely used method for in vitro multiplication is propagation from preexisting meristems using shoot tips and nodal segments [26]. Our results show that the more effective explants are nodal segments rather than shoot tips. This finding is consistent with data in the literature showing that nodal explants possess high regeneration potential [27]. Shoot tip explants exhibit strong apical dominance, which inhibits bud sprouting and shoot proliferation [28]. Some authors, however, report that the use of shoot tip explants results in significant shoot multiplication [29,30]. The observed differences in the regenerative capacity of the various explants may be due to differences in the endogenous hormones and nutrients in the plant tissues [31]. The success and cost-effectiveness of the micropropagation protocol depend on high in vitro rooting rates and plant survival during the transfer of plants from culture to field conditions, leading to large-scale production [32]. In the current study, all shoots were simultaneously rooted on half-strength medium without any PGR supplementation. The plantlets were successfully adapted to ex vitro conditions with a high survival rate. The acclimatized plants exhibited normal growth and true-to-type morphology under outdoor conditions. 

Polyphenols and flavonoids are plant constituents produced as an adaptive response to different biotic and abiotic conditions, and their contents may vary depending on environmental factors such as the photoperiod, altitude, UV radiation level, humidity and temperature [33,34]. They deactivate free radicals, degrade peroxides, inactivate metals and scavenge oxygen, thus preventing oxidative damage in plants [35,36,37]. The total polyphenol content and antioxidant activity of wild *C. vulgare* extracts have been reported in previous publications [38,39,40,41,42], but to our knowledge, information on the polyphenol and flavonoid contents, as well as the antioxidant activity, of in vitro cultivated *C. vulgare* plants is lacking. The total phenolic content (225.37 mg GAE/g) of *C. vulgare* aqueous extracts found by Balkan et al. [40] was similar to ours, whereas values reported by Bektašević et al. [41] and Vladimir-Knežević et al. [42] were lower, with those of Bektašević et al. [41] being much lower. 

In this study, differences in the total polyphenol and flavonoid contents, as well as the antioxidant activity of different organs—leaves, flowers and stems—of wild and cultivated *C. vulgare* plants were found. Regarding wild-growing *C. vulgare* plants, compared to flowers and stems, leaves were richer in both polyphenols and flavonoids. Regarding cultivated plants, flowers were the richest in polyphenols, whereas leaves were the richest in flavonoids. The content of polyphenols in the flowers of cultivated plants was significantly higher than that in the flowers of wild-growing plants. The same held true for the antioxidant activity of flowers from in vitro plants compared to flowers of wild plants, with the ORAC activity of the flowers of in vitro grown plants being higher than that of the flowers of wild plants, making the flowers of cultivated plants a suitable candidate as a source of biologically active substances with significant antioxidant activity. To the best of our knowledge, this is the first report on the evaluation of the ORAC and HORAC antioxidant activities of in vitro cultivated and wild-growing *C. vulgare* plants. The ORAC method assesses the peroxyl radical chain-breaking ability of antioxidants via the hydrogen atom transfer pathway, while the HORAC method measures the metal-chelating radical prevention activity of the sample. The high antioxidant activity of leaves could possibly be explained by the high contents of phenolic acids such as chlorogenic, rosmarinic and caffeic acids. The phenolic compounds we detected in the samples have been reported by other authors as well. For example, ethanolic extracts of *C. vulgare* were shown to contain rosmarinic acid (12.27 mg/g), chlorogenic acid (2.25 mg/g) and caffeic acid (0.56 mg/g) [42], whereas the water extracts contained catechin, epicatechin, rutin and apigenin [43]. Bektašević et al. [41] investigated the phenolic compounds in freeze-dried aqueous and methanolic *C. vulgare* extracts and revealed the main phenolic compounds to be rosmarinic acid (26.63 and 34.2), ellagic acid (23.11 and 29.31), ferulic acid (3.99 and 2.19), p-coumaric acid (2.53 and 1.94) and myricetin (0.83 and 1.09) (mg/g of extract). Amirova et al. [11] also observed that the extracts of *C. vulgare* contained the same phenolic acids (caffeic, chlorogenic and rosmarinic acids) and flavonoids (apigenin, kaempferol and catechin). However, unlike our study, most of these reports do not distinguish between the different plant parts of *C. vulgare*.

Subsequently, the antitumor activity of the extracts of cultivated plants was assessed and compared with that of the extracts of wild plants. While leaf extracts at a dosage of 500 µg/mL, from both cultivated and wild plants, consistently showed the best cytotoxic activity, especially in the HeLa and HT-29 cell lines, our attention was attracted to the effect of the extracts of cultivated leaves (250 µg/mL) and cultivated flowers (500 µg/mL) because of the lack of a significant cytotoxic effect of the latter on the healthy keratinocyte culture (used as a control). Our results showed that, even at a low dosage of 250 µg/mL, the cultivated plants’ leaf extract significantly lowered the number of viable cells in the HT-29 cell line at 24 h and in the HT-29 and MCF-7 cell lines at 48 h while having no significant detrimental effect on the healthy control line (human keratinocyte cell line HaCaT). The same applied to the extract of cultivated flowers at a dosage of 500 µg/mL for the HT-29 and HeLa cell lines, but also to the MCF-7 cell lines at both 24 and 48 h. We believe that the beneficial effects of the two extracts could be due to the high contents of neochlorogenic acid and catechin or rather the combination of the two. Indeed, the highest content of neochlorogenic acid was found in extracts from cultivated flowers (the highest amount, standing out with a value of 790.6 ± 1.0) and cultivated leaves (595.6 ± 1.7), and catechin was found only in the leaves and flowers of cultivated plants while being absent in the wild plants and stems of cultivated plants. Neochlorogenic acid (Neochlorogenate, 3-O-Caffeoylquinic acid), a less-studied isomer of chlorogenic acid, has been detected in some fruits (peaches, prunes, plums, coffee beans, apricots and cherries) and herbs (*Geranium purpureum, Crataegus monogyna* and rosemary leaves) [44] and shown to possesses multiple pharmacological activities, including antioxidant, antifungal, anti-inflammatory and antitumor effects [45,46,47,48,49,50,51]. It has been demonstrated that neochlorogenic acid possesses strong anticancer activity against several types of human cancer [52,53]. It effectively reduces the growth and volume of tumors induced by ACS gastric cancer cells injected in nude mice, with the underlying mechanism being the induction of apoptosis, reactive oxygen species (ROS) formation, the loss of mitochondrial membrane potential (MMP), m-TOR/PI3K/AKT signaling inhibition and the withholding of cancer cell migration and invasion. Furthermore, *Leonurus sibiricus* extracts, rich in neochlorogenic acid, showed anticancer effects against grade IV human glioma cells and the U87MG cell line related to DNA damage and the downregulation of several epigenetic factors in the cancer cells [53]. Catechins, in turn, have been shown to inhibit tumorigenesis, tumor growth, cancer cell invasion and tumor angiogenesis by inhibiting the induction of proangiogenic factors [54]. Several in vitro cell experiments and *in vivo* animal experiments have shown that catechins have a strong anticarcinogenic effect and effectively suppress the metastasis and invasion of various cancer cells [55]. It should be noted that the flower extracts of wild plants also show good cytotoxic activity against cancer cells, with no significant damaging effects on control cells, especially at 24 h, but the best ratio (cancer cell cytotoxic activity/no harm to healthy cells) remains attributed to the cultivated flower extract at 48 h at a dosage of 500 µg/mL for the HT-29 and HeLa tumor cell lines. Data on the antitumor activity of in vitro cultivated *C. vulgare* plants are not available, and our study is the first to show the antitumor activity of in vitro cultivated plants. A couple of reports demonstrate antitumor activity for wild-growing plants [19]. Batsalova et al. [2] investigated the effects of three types of extracts (acidified, alkalized and lipophilic) on CaOV (human testis cystadenocarcinoma), HeLa and HT-29 cell lines. Dzhambazov et al. [19] studied the influence of aqueous extracts on cells from the human larynx epidermoid carcinoma (HEp-2), human metastatic melanoma cells (A2058) and mouse lymphoma cells (L5178Y). Similar to our study, Batsalova et al. [2] reported toxic activity on cancer cells from HeLa and HT-29 cell lines, but, in their study, they did not use aqueous extracts, and differences in cytotoxic activity are likely; thus, direct comparisons are not possible. 

## 4. Materials and Methods

### 4.1. In Vitro Culture 

#### 4.1.1. Initial Plant Material 

Seeds were collected from wild plants from the local population in the Vitosha Mountains, near the village of Bistritsa (Sofia region, Bulgaria), in open grassland with shrubs, 900 m altitude, Bulgaria, and used as the initial in vitro plant material. Explants for in vitro propagation were obtained from two-month-old seedlings using two types of explants: (1) shoot tips (6–8 mm in size) and (2) nodal segments (8–10 mm in size) with a leaf node and two adjacent leaves.

#### 4.1.2. Seed Sterilization 

Seeds were surface-sterilized with 70% ethyl alcohol for 2 min and then soaked in 50% commercial bleach (containing 4.85% active chlorine) for 15 min. Before sterilization, seeds were washed with running tap water and detergent, and then the disinfection scheme was applied. A drop of Twin-20 emulsifier (Sigma-Aldrich, St. Louis, MO, USA) was added to the decontamination solution. Each procedure was followed by a triple rinse for 5, 10 and 15 min in autoclaved distilled water. Treated seeds were placed in 9 cm Petri dishes.

#### 4.1.3. Media Composition for In Vitro Cultivation

Seeds were germinated on Murashige and Skoog basal medium (MS) [20] free of PGRs. For shoot proliferation, explants (shoot tips and nodal segments) were cultivated on full-strength MS medium supplemented with one of the following cytokinins: 6-benzylaminopurine (BAP), kinetin, zeatin or N6 [2-isopentenyl]-adenine (2-iP) at a concentration of 1 mg/L alone or in combination with the auxin indole-3-butyric acid (IBA) at a concentration of 0.1 mg/L. The composition of the tested nutrient media was as follows (Table 1): B1 (1 mg/L BAP); iP1 (1 mg/L 2-iP); Z1 (1 mg/L zeatin); K1 (1 mg/L kinetin); B1I0.1 (1 mg/L BAP and 0.1 mg/L IBA); iP1I0.1 (2-iP and 0.1 mg/L IBA); Z1I0.1 (1 mg/L zeatin; and 0.1 mg/L IBA) K1I0.1 (1 mg/L kinetin and 0.1 mg/L IBA).

Every 4 weeks, the shoots were separated and subcultivated using the shoot tips and the nodal segments of the new (newly formed) shoots. 

#### 4.1.4. In Vitro Rooting and Acclimatization of Obtained Plants

For root induction, shoots were cultivated on half-strength MS-based medium without PGRs for four weeks. Rooted plants were removed from culture tubes and washed with running tap water to remove traces of agar. Afterward, plantlets were transferred to plastic pots (8 cm in diameter) containing a mixture of soil, peat, perlite and sand in a ratio of 2:1:1:1 (*v*/*v*/*v*/*v*). For two weeks, the pots were covered with clear plastic boxes to maintain high relative humidity. Plastic boxes were then gradually removed for the ex vitro adaptation of the plants. The survival rate was recorded for 5 weeks after adaptation under ex vitro conditions. The obtained plants were then moved to the greenhouse for acclimatization and finally planted outdoors in Elin Pelin field plots at 640 m altitude in April 2021.

All tested media contained 3% sucrose and were solidified with 0.7% agar (*w*/*v*) (Duchefa). The pH of each medium was adjusted to 5.8 using sodium hydroxide (NaOH) and hydrogen chloride (HCl). All media used for plant culturing were autoclaved at 1.1 kg/cm^2^ at 121 °C for 20 min.

The frequency of shoot proliferation (evaluated as % of explants forming shoots), the multiplication rate (evaluated as the mean number of shoots per explant) and the mean shoot height were measured after four weeks of incubation. Subculturing was performed at four-week intervals. Each experimental variant included 40 explants in duplicate. 

#### 4.1.5. Conditions for In Vitro Cultures

In vitro materials were cultivated in a growth room with artificial illumination (fluorescent lamps type FL-40 W^−1^, Svetlina Ltd., Stara Zagora, Bulgaria) and a 16 h photoperiod at 18–21 °C, with a photon flux density of 40 μM m^−2^ s^−1^.

### 4.2. Phytochemical Analysis

#### 4.2.1. Plant Material

For phytochemical and antitumor activity analyses, the aerial parts of wild plants as well as the aerial parts of cultivated in vitro plants grown in the experimental field plot were collected. The plant materials were collected in the summer (June) of 2022 and shade-dried at room temperature. Leaves, flowers and stems were separated, packed in paper bags and stored at room temperature before analysis.

#### 4.2.2. Preparation of Freeze-Dried Extracts

Approximately 50 g of dried plant mass (stems, leaves and flowers) was ground in a laboratory grinder to a fine powder. For the extraction, 5 g of the powder was added to 200 mL of water (90 °C) and incubated for 15 min. The suspension was further centrifuged at 6000× *g*, and the supernatant was collected and freeze-dried for 96 h in an Alpha 1–4 LD plus laboratory freeze dryer (Martin Christ Gefriertrocknungsanlagen GmbH, Osterode am Harz, Germany). Based on the used plant part, dry extracts were denoted as follows:FCP—flowers of in vitro cultivated plants;LCP—leaves of in vitro cultivated plants;SCP—stem of in vitro cultivated plants;FWP—flowers of wild plants;LWP—leaves of wild plants;SWP—stems of wild plants.

#### 4.2.3. Total Polyphenol and Total Flavonoid Contents

Total polyphenols were determined according to the method of Singleton and Rossi [56] with the Folin–Ciocalteu reagent. Gallic acid was employed as a calibration standard, and the results are expressed in mg gallic acid equivalents (GAE) per 100 g dry weight (DW) ± SD.

The total flavonoid content was determined with aluminum chloride (AlCl_3_) reagent according to Chang et al. [57]. The calibration curve was constructed using quercetin dihydrate (10–200 mg/L). The results are expressed as mg quercetin equivalents (QE) per 100 g DW ± SD.

#### 4.2.4. Antioxidant Activity Assays

Oxygen radical absorbance capacity (ORAC) activity was measured on a microplate reader (FLUOstar OPTIMA; BMG Labtech, Ortenberg, Germany) with excitation at λ = 485 nm and emission at λ = 520 nm, according to the method of Ou et al. [58], with some modifications described by Denev et al. [59]. Fluorescein (FL) served as the fluorescent probe, and trolox was used as a standard antioxidant to generate the standard curve. The radical scavenging properties of the freeze-dried extracts were studied by comparing the area under the fluorescence curve relative to this area in the absence of a sample (control). The results are expressed in micromole trolox equivalents (μmol TE) per gram DW ± SD. 

The hydroxyl radical averting capacity (HORAC) activity of the freeze-dried extracts was determined with excitation at λ = 485 nm and emission at λ = 520 nm, according to Ou et al. [58]. FL was used as a fluorescent agent, and gallic acid served as a standard. The protective effect of the samples was determined as explained for the ORAC method. The results are expressed in micromole gallic acid equivalents (μmol GAE) per gram DW ± SD.

#### 4.2.5. HPLC Determination of Phenolic and Flavonoid Compounds

HPLC analyses were performed on a UHPLC system (Nexera-i LC2040C Plus; Shimadzu Corporation, Kyoto, Japan) with a UV-VIS detector and a binary pump. The column was Poroshell 120 EC-C18 (3 mm × 100 mm, 2.7 μm), thermostated at 26 °C. The flow rate was 0.3 mL/min, and the injection volume was 5 μL. The detection of the derivatives was performed at λ = 280 nm. The mobile phase consisted of 0.5% acetic acid (A) and 100% acetonitrile (B). The gradient condition started with 14% (B), and between 6 and 30 min, it linearly increased to 25% (B) and then increased to 50% (B) at 40 min. The identification of compounds was carried out by comparing retention times utilizing standard solutions and standard calibration curves for different phenolics. The results for individual phenolic compounds are expressed in mg per 100 g dry weight (DW) ± SD.

### 4.3. Antitumor Activity

#### 4.3.1. Cell Lines

The human tumor cell lines HeLa (cervical carcinoma), HT-29 (colorectal carcinoma) and MCF-7 (mammary carcinoma) were obtained from the American Type Culture Collection (ATCC), and the HaCaT cell line (human keratinocytes) was obtained from the CLS Cell Lines Service (Eppelheim, Germany). Cells were cultivated in 25 cm^2^ tissue culture flasks in Dulbecco’s modified Eagle medium (Gibco) supplemented with 10% fetal bovine serum, 2 mM glutamine and the antibiotics penicillin 100 U mL^−1^ and streptomycin 100 µg mL^−1^ at 37 °C, 5% CO_2_ and 90% relative humidity.

#### 4.3.2. Test Sample Preparation

The lyophilized extracts were dissolved in Dulbecco’s modified Eagle medium (DMEM) at a concentration of 10 mg/mL and sterilized by filtration through 0.2 μm pore size syringe filters. Extracts were further diluted to the desired concentrations with DMEM containing 10% fetal bovine serum.

#### 4.3.3. Cell Viability Assay

The effects of *C. vulgare* extracts on the viability of tumor cells were assessed using the MTT assay. Briefly, cells were trypsinized with 0.25 % Trypsin-EDTA and counted using a hemocytometer. Cells were then transferred to a 96-well microtiter plate to ensure a concentration of 1 × 104 cells/well in the medium with a final volume of 100 μL and incubated overnight at 37 °C in a humidified atmosphere containing 5% CO_2_ to allow cell attachment. Subsequently, cells were treated with four different concentrations of the *C. vulgare* extracts (125 μg/mL, 250 μg/mL, 500 μg/mL and 1000 μg/mL) and incubated for an additional 24 and 48 h. Tumor cells cultivated in the culture medium alone were used as controls. To evaluate the cytotoxic activity of the tested extracts on non-tumor cells, parallel experiments with the same treatment and incubation regimen were performed on HaCaT human keratinocyte cells. Each extract was tested through five measurements. After culturing in the presence of extracts, cells were washed twice with PBS (pH 7.4) and further incubated with 100 μL of the MTT working solution (Sigma Chemical) at 37 °C for 3 h; the supernatants were aspirated, and 100 μL of a lysing solution (DMSO/ethanol 1:1) was added to each well to dissolve the resulting formazan. MTT assay readings were taken using an ELISA plate reader (TECAN, SunriseTM, Grodig/Salzburg, Austria). The cell viability of the treated cells is presented as a percentage of the untreated control.

#### 4.3.4. Fluorescence Microscopy Analysis

The morphological alterations induced in cancer cells by the extract of flowers of in vitro cultivated *C. vulgare* were analyzed by fluorescence microscopy. For this purpose, cells were cultivated on 13 mm diameter cover glasses in 24-well plates and treated for 24 h with the extract at a concentration equal to the IC_50_ value determined by the MTT test. Two different fluorescent staining methods were used for the cytomorphological analysis of extract-treated tumor cells. The ability of the tested extract to induce cytotoxic damage in the tumor cells was assessed by live/dead double staining with acridine orange (AO) and ethidium bromide (EtBr). For this purpose, native cell preparations of tumor cells treated with *C. vulgare* extracts were stained with the fluorescent dyes AO (5 µg mL^−1^) and EtBr (5 μg mL^−1^) in PBS.

The alterations in the nuclear morphology of tumor cells induced by the *C. vulgare* extracts were studied after staining with the DNA-binding dye 4′,6-diamidine-2′-phenylindole dihydrochloride (DAPI). Cells were fixed with methanol and incubated for 15 min in 1 µg mL^−1^ DAPI in methanol in the dark. 

Stained cell cultures were visualized and examined under a fluorescence microscope (Leica DM 5000B, Wetzlar, Germany).

### 4.4. Statistical Analysis

Data from in vitro propagation experiments were subjected to one-way ANOVA (analysis of variance) for the comparison of means, and significant differences were calculated according to Fisher’s least significance difference (LSD) test at a 5% significance level using a statistical software package (Statgraphics Plus, version 5.1 for Windows). Data are presented as means ± standard deviations (SDs). All experiments were repeated three times. The results of phytochemical analyses and biological activity tests were calculated using Microsoft Excel (Microsoft Corporation). Data are expressed as means ± standard deviations (SDs). All experiments were repeated four times. For antitumor activity, the statistical significance of data was assessed by one-way analysis of variance (ANOVA) followed by Bonferroni’s post hoc test using GraphPad PRISM software. Nonlinear regression (curve fit analysis) was used to determine IC_50_ concentrations.

## 5. Conclusions

In the present study, a protocol for the in vitro propagation of *C. vulgare* with the high survival of adapted plantlets was developed. For the first time, the chemical content and antioxidant and antitumor activities of cultivated plants were studied and compared with those of wild-type plants. In vitro cultivated plants showed strong antitumor activity, most likely due to the high contents of detected neochlorogenic acid and catechin, known for their antitumor effects. 

## Figures and Tables

**Figure 1 plants-12-01591-f001:**
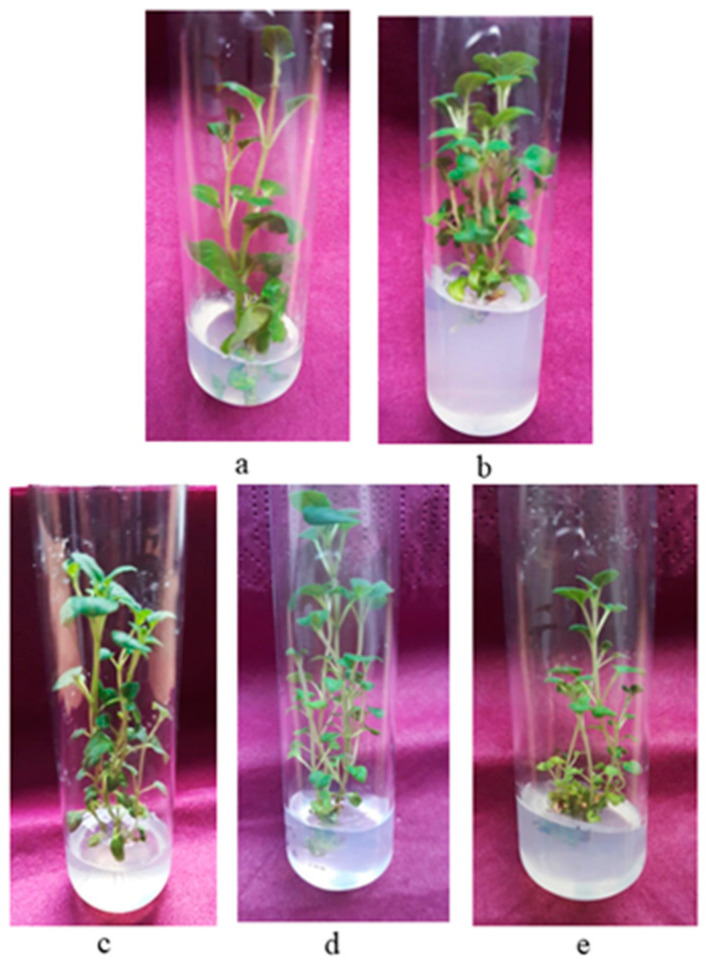
Micropropagation of *Clinopodium vulgare* L. on MS medium with different PGRs: (**a**) control MS medium, (**b**) B1, (**c**) Z1, (**d**) K1 and (**e**) iP1.

**Figure 2 plants-12-01591-f002:**
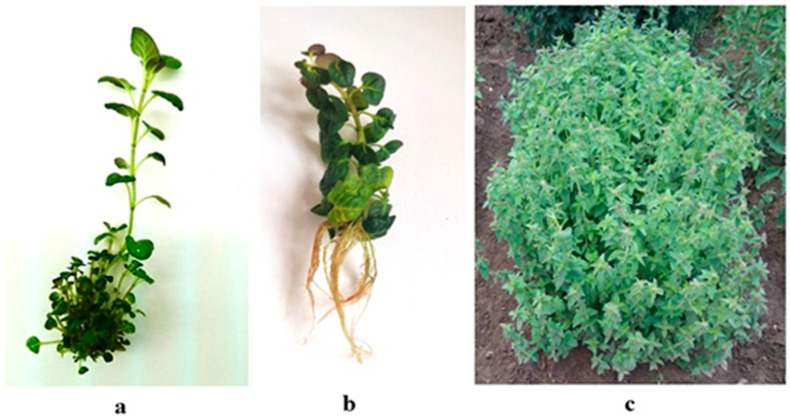
In vitro cultivation of *Clinopodium vulgare* L.: (**a**) shoot multiplication on MS medium supplemented with B1I0.1, (**b**) in vitro rooting on half-strength MS medium and (**c**) plants cultivated in the experimental field.

**Figure 3 plants-12-01591-f003:**
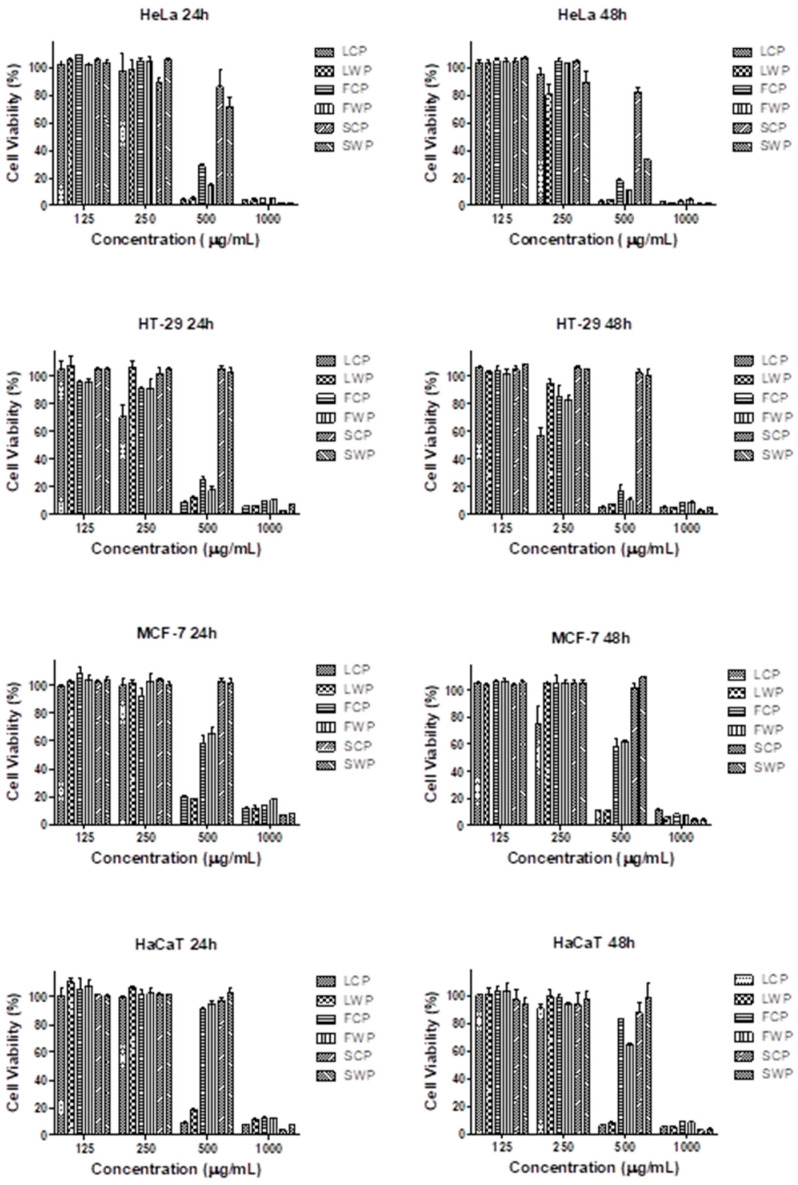
Effects of *C. vulgare* extracts on the cell viability of HeLa, HT-29, MCF-7 and HaCaT cell lines as assessed by MTT test after 24 and 48 h of exposure.

**Figure 4 plants-12-01591-f004:**
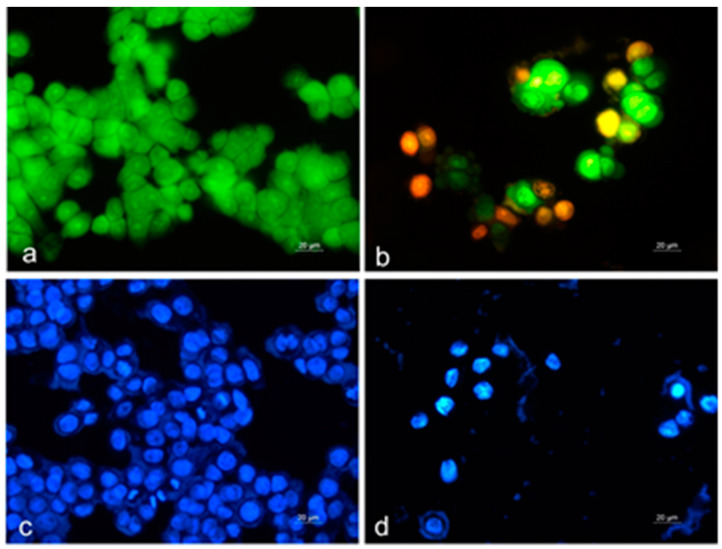
Cytomorphological alterations induced by the extract of in vitro cultures of *C. vulgare* in HT-29 colorectal carcinoma cells. (**a**,**c**) Control untreated cells; (**b**,**d**) cells treated with *C. vulgare* extract. (**a**,**b**) AO/EB staining; (**c**,**d**) DAPI staining; fluorescence microscopy; objective 40×.

**Table 1 plants-12-01591-t001:** Effect of plant growth regulators on micropropagation of *Clinopodium vulgare*—I^st^ subculture.

Nutrient Medium	Shoot Tips			Nodal Segments		
	Shoot Formation (%)	Mean Number of Shoots Per Explant	Mean Height of Shoots (cm)	Shoot Formation (%)	Mean Number of Shoots Per Explants	Mean Height of Shoots (cm)
MS	10	1.15 ± 0.36 ^a^	7.55 ± 1.24 ^e^	10	1.2 ± 0.41 ^a^	8.3 ± 1.54 ^d^
B1	60	2.05 ± 0.68 ^cd^	3.85 ± 1.00 ^c^	90	4.6 ± 1.69 ^e^	4.78 ± 0.70 ^b^
iP1	15	1.35 ± 0.48 ^ab^	3.95 ± 1.02 ^c^	30	2.15 ± 0.67 ^b^	3.45 ± 1.05 ^a^
Z1	40	1.95 ± 0.60 ^cd^	3.1 ± 0.80 ^ab^	70	3.4 ± 1.09 ^cd^	5.35 ± 1.59 ^bc^
K1	30	1.75 ± 0.55 ^bc^	5.85 ± 1.14 ^d^	80	2.65 ± 0.81 ^bc^	6.0 ± 1.12 ^c^
B1I0.1	80	3.0 ± 0.72 ^e^	3.6 ± 1.04 ^bc^	100	6.9 ± 2.57 ^f^	4.65 ± 0.86 ^b^
iP1I0.1	20	1.4 ± 0.59 ^ab^	2.9 ± 0.92 ^a^	100	2.6 ± 0.82 ^bc^	3.7 ± 0.76 ^a^
Z1I0.1	60	2.3 ± 0.57 ^d^	2.86 ± 0.78 ^a^	100	4.0 ± 1.55 ^de^	5.23 ± 2.36 ^bc^
K1I0.1	40	1.9 ± 0.64 ^c^	4.1 ± 0.89 ^c^	100	3.6 ± 1.18 ^d^	4.8 ± 1.42 ^b^
LSD		0.37	0.62		0.84	0.85

The data are presented as means of 40 shoots per medium variant ± standard deviation (SD). Different letters indicate significant differences assessed by Fisher LSD test (*p* ≤ 0.05) after performing one-way ANOVA.

**Table 2 plants-12-01591-t002:** Total polyphenol and flavonoid contents and antioxidant activity of freeze-dried aqueous extracts from different anatomical parts of in vitro cultivated and wild-growing *C. vulgare* plants.

Samples	Total Polyphenols, mg GAE/100 g DW	Total Flavonoids, mg QE/100 g DW	ORAC, µmol TE/g DW	HORAC, µmol GAE/g DW
FCP	29,927.6 ± 592.1 ^d^	763.2 ± 37.2 ^a^	7281.3 ± 82.9 ^c^	2251.0 ± 21.2 ^c^
LCP	28,016.9 ± 447.3 ^c^	1100.9 ± 81.5 ^b^	7342.5 ± 54.4 ^c^	2326.9 ± 56.0 ^d^
SCP	16,733.7 ± 390.0 ^a^	824.6 ± 109.1 ^a^	3393.1 ± 40.2 ^a^	1278.8 ± 32.4 ^b^
FWP	27,292.8 ± 85.3 ^b^	1460.5 ± 63.1 ^c^	7246.3 ± 62.4 ^c^	2534.0 ± 45.8 ^e^
LWP	30,510.9 ± 85.3 ^d^	1815.8 ± 91.2 ^d^	7258.2 ± 105.9 ^c^	2572.7 ± 41.8 ^e^
SWP	16,713.6 ± 142.2 ^a^	1614.0 ± 154.3 ^c^	3849.9 ± 74.0 ^b^	1179.4 ± 18.3 ^a^
LSD	623.72	171.92	129.98	68.18

Legend: FCP (flowers of in vitro cultivated plants); LCP (leaves of in vitro cultivated plants); SCP (stems of in vitro cultivated plants); FWP (flowers of wild plants); LCP (leaves of wild plants); SCP (stems of wild plants). The data are presented as means of 3 samples ± standard deviation. Different letters indicate significant differences assessed by Fisher LSD test (*p* ≤ 0.05) after performing multifactor ANOVA.

**Table 3 plants-12-01591-t003:** Phenolic constituents (mg/100 g DW) of in vitro cultivated and wild-growing *C. vulgare* plants.

Samples	Neochlorogenic Acid	Chlorogenic Acid	Catechin	Rosmarinic Acid	Quercetin	Apigenin	Caffeic Acid	Sum
FCP	790.6 ± 1.0 ^f^	52.7 ± 0.4 ^a^	222.9 ± 5.3 ^b^	369.3 ± 0.3 ^c^	-	-	65.9 ± 0.9 ^c^	1501.4
LCP	595.6 ± 1.7 ^e^	183.0 ± 0.8 ^d^	136.6 ± 6.1 ^a^	1286.6 ± 1.8 ^e^	-	-	23.0 ± 0.6 ^a^	2224.8
SCP	244.8 ± 0.2 ^b^	175.0 ± 1.3 ^c^		114.8 ± 0.8 ^b^	565.8 ± 5.6 ^b^	125.3 ± 0.2 ^b^	23.0 ± 0.9 ^a^	1248.7
FWP	526.8 ± 0.2 ^c^	81.1 ± 1.6 ^b^	-	851.8 ± 0.2 ^d^	-	-	69.3 ± 0.9 ^d^	1529.0
LWP	575.0 ± 0.1 ^d^	205.6 ± 0.3 ^e^	-	1734.6 ± 4.2 ^f^	-	-	31.1 ± 0.4 ^b^	2546.2
SCP	193.0 ± 1.3 ^a^	259.5 ± 0.2 ^f^	-	87.5 ± 1.6 ^a^	319.7 ± 0.5 ^a^	39.3 ± 0.3 ^a^	31.1 ± 0.1 ^b^	930.1
LSD	1.73	1.65	12.95	3.57	9.01	0.57	1.25	

Legend: FCP (flowers of in vitro cultivated plants); LCP (leaves of in vitro cultivated plants); SCP (stems of in vitro cultivated plants); FWP (flowers of wild plants); LCP (leaves of wild plants); SCP (stems of wild plants). The data are presented as means of 3 samples ± standard deviation. Different letters indicate significant differences assessed by Fisher LSD test (*p* ≤ 0.05) after performing multifactor ANOVA.

**Table 4 plants-12-01591-t004:** Half-maximal inhibitory concentrations (IC_50_) of extracts from in vitro cultivated and wild-growing *C. vulgare*.

IC_50_	HeLa	HeLa	HT-29	HT-29	MCF-7	MCF-7	HaCaT	HaCaT
(µg/mL)	24 h	48 h	24 h	48 h	24 h	48 h	24 h	48 h
LCP	361.6	343.9	304.4	266.5	429.0	319.1	400.0	345.1
LWP	380.8	309.0	426.8	363.0	427.8	426.3	427.6	398.9
FCP	479.6	467.8	398.9	362.0	557.0	537.8	729.1	659.6
FWP	464.2	457.9	380.9	336.5	620.3	556.9	753.9	557.0
SCP	621.2	597.1	882.5	880.1	933.7	924.7	721.7	639.7
SWP	560.0	423.6	934.1	928.8	938.2	924.7	906.4	742.4

## Data Availability

All data is comprised in the manuscript.

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
