# Peer review of "Antitumor and Antioxidant Activities of In Vitro Cultivated and Wild-Growing *Clinopodium vulgare* L. Plants"

_plants, 2023, doi:10.3390/plants12081591_

Round 1
Reviewer 1 Report
All suggestions are included in the manuscript text at the appropriate places.

Author Response
First of all, we would like to thank you for your valuable comments. We greatly appreciate your precious comments for improving the quality of our manuscript. We are sure that you could find the revised version of the manuscript more accurate.
Page 1 line 2: Please correct antitumour to antitumor throughout the text
Response to reviewer: The word was corrected throughout the text.
Page 1 line 29: Cultivated or cultured? Please choose terminology and use it consistently throughout the text. In my opinion cultivated is more appropriate term.
Response to reviewer: The term cultivated was used instead of cultured throughout the text as recommended by the reviewer.
Page 1 line 31: the
Response to reviewer: The definite article “The” was added.
Page 1 line 32: the
Response to reviewer: The definite article “the” was added.
Page 1 line 34: compounds
Response to reviewer: The term “compounds” was used instead “substances”
Page 5 line 140: s
The content was corrected to contents
Page 5 line 142: “mg equivalents GA or mg GAE”
Response to reviewer: the equivalents “GAE” was added
Page 5 line 149: This part of the text should be included in a separate subsection called “Antioxidant activity”. Add (Table 2) in the end of this paragraph. Although polyphenolic content is closely related to antioxidant activity, this part of the text describes biological activity.
Response to reviewer: We agree with the reviewer that the antioxidant activity deserves special attention, differentiating it from the chemical composition of extracts and that's why we have placed it in a separate paragraph that follows the chemical composition.
The separate subsection called “Antioxidant activity” was added. The Table 2 was moved in the end of the paragraph.
Page 8 line 209: “Use subscript IC50”
Response to reviewer: The subscript “IC50” was used
Page 9 line 248: do not
Response to reviewer: We used “do not” instead don’t
Page 9 line 251: reported
Response to reviewer: We used “reported” instead “report”
Page 10 line 299: italic
Response to reviewer: The “C. vulgare” was italicized.
Page 10 line 307: Not clear, rephrase
Response to reviewer: The sentence is written more clearly.
Regarding cultivated plants, flowers were the richest in polyphenols, whereas leaves - in flavonoids. The content of polyphenols in the flowers of cultivated plants, was significantly higher than that in the flowers of wild growing plants.
Page 11 line 313: the significant
Response to reviewer: “the significant” was added.
Page 11 line 313: “??? You mentioned this data for the first time in the text. It is not necessary.”
Response to reviewer: “lyophilized extracts” was deleted.
Page 11 line 325: phenolic compounds
Response to reviewer: The “phenolic compounds” was used instead “representatives”
Page 11 Line 331:
All mentioned are different plant parts (plant organs), not anatomical parts
Response to reviewer: We used plant parts instead anatomical parts.
Page 11 line 332:
Response to reviewer: The comma was added
Page 11 line 334: the, delete antitumor
Response to reviewer: The definite article “the” was added. The word antitumor was deleted.
Page 11 line 339: plants'
Response to reviewer: The word “plants'” was added .
Page 11 line 385:
Note the voucher number or seed bank entry for your sample.
Response to reviewer: Seeds were collected from wild plants of the local population in the Vitosha Mountains, near the village of Bistritsa (Sofia region, Bulgaria), in an open grassland with shrubs, 900 m altitude, Bulgaria, and used as initial in vitro plant material
The wild growing plants from which the seeds were collected were identified by taxonomist A/Prof Dr. Ina Aneva (ORCID: 0000-0002-6476-5438) from the Institute of Biodiversity and Ecosystem Research, Bulgarian Academy of Sciences.
Page 12 line 412: and line 416 italic
Response to reviewer: The “Ex vitro” was italicized.
Page 13 line 422:
Write compounds names instead formulas
Response to reviewer: The compounds names were written instead formulas
Page 13 line 436: See above comment. Choose the most appropriate term and use it consistently throughout the text
Response to reviewer: The term cultivated was used instead acclimatized
Page 13 line 446: plant parts
Response to reviewer: The plant parts were used instead anatomical parts.
Page 13 line 460: Aluminum chloride instead formula
Response to reviewer: The Aluminum chloride was written instead formula.
Page 13 line 461: Constructed
Response to reviewer: The word “built” was replaced with “constructed”.
Page 14 line 496: Delete space
Response to reviewer: The space was deleted.
Page 14 line 508: Add space
Response to reviewer: The space was added.
Rather use L instead l
Response to reviewer: The “l” was replaced with “L”.
Page 14 line 509: Delete space
Response to reviewer: The space was deleted.
Page 14 line 518: Add space
Response to reviewer: The space was added.
Page 15 line 531: Add space
The space was added.
Page 15 line 533: Italic
Response to reviewer: The “C. vulgare” was italicized.
Page 15 line 539: Statistical analysis
Response to reviewer: We used “Statistical analysis” instead Statistics
Page 15 line 545:
Response to reviewer: The sentence “Phytochemical analyses were performed using Microsoft Excel (Microsoft Corporation)” was replaced with “The results of phytochemical analysis and biological activity test were calculated using Microsoft Excel”
Page 15 line 560:
Delete marked text from the template
Response to reviewer: The marked text was deleted from the template.
Reviewer 2 Report
This study shows an efficient protocol for the micropropagation of c. Vulgare and compares some biological properties and contents but they need to describe more the hplc results in a table or in supplementary doc.
Author Response
We would like to thank the reviewer for his work and are grateful for the high appreciation of our manuscript.
Review Report: “This study shows an efficient protocol for the micropropagation of c. Vulgare and compares some biological properties and contents but they need to describe more the hplc results in a table or in supplementary doc.”
Reviewer Response 2: Thank you for appreciating our work. In fact, we dedicated a whole separate table to HPLC results. It is Table 3 (page 6) and results are discussed in detail on page 11. We believe that results from HPLC analysis are well illustrated and properly discussed in line with the conclusions of the study.
Reviewer 3 Report
The paper described a protocol for micropropagation of Clinopodium vulgare and compared the chemical composition, the relative content of compounds, as well as antitumour and antioxidant activities of extracts from in vitro cultured and wild growing plants.
A minor correction should be done in the paper: neochlorogenic acid is not the 5-caffeoylquinic acid. Actually, the literature has a great confusion about the designation of 3-O-caffeoylquinic acid and 5-O-caffeoylquinic acid as chlorogenic acid. The nomenclature of chlorogenic acid derivatives was revised in a review paper published by Clifford et al., 2017 [Nat. Prod. Rep., 2017, 34, 1391-1421 (http://dx.doi.org/10.1039/C7NP00030H)] and J. Nat. Prod.2017, 80, 1028–1033 (https://doi.org/10.1021/acs.jafc.7b00729)].
Author Response
We would to thank the referee for the valuable comments that we believe would improve the quality of the manuscript.
Review Report: “The paper described a protocol for micropropagation of Clinopodium vulgare and compared the chemical composition, the relative content of compounds, as well as antitumour and antioxidant activities of extracts from in vitro cultured and wild growing plants.
A minor correction should be done in the paper: neochlorogenic acid is not the 5-caffeoylquinic acid. Actually, the literature has a great confusion about the designation of 3-O-caffeoylquinic acid and 5-O-caffeoylquinic acid as chlorogenic acid. The nomenclature of chlorogenic acid derivatives was revised in a review paper published by Clifford et al., 2017 [Nat. Prod. Rep., 2017, 34, 1391-1421 (http://dx.doi.org/10.1039/C7NP00030H)] and J. Nat. Prod.2017, 80, 1028–1033 (https://doi.org/10.1021/acs.jafc.7b00729)].”
Response to reviewer: Thank you for appreciating our work and for the comment. Indeed, there is a big confusion in the literature about the nomenclature of chlorogenic acid derivatives and many papers are being submitted and published designating erroneously chlorogenic and neochlorogenic acids. We revised the name of neochlorogenic acid from 5-O-Caffeoylquinic acid to 3-O-Caffeoylquinic acid.